# OpenReview forum: "The Illusion of Role Separation: Hidden Shortcuts in LLM Role Learning (and How to Fix Them)"
_ICML.cc/2025/Conference — ICML 2025 poster_

### Official Review · Reviewer_r6iR · 2025-03-09

**Overall Recommendation:** 4

**Summary:**

The paper studies issues in LM role-learning for security purposes (e.g., following instructions in system prompts over user instructions), and identifies two key issues: task type exploitation (the model following user tasks that are similar to the system prompts) and proximity to beginning of text (the model follows instructions close to the start of the input). The authors propose two methods for addressing these, finetuning on swapped user and system roles (for task type exploitation) and “PFT” (position-enhanced finetuning; for proximity to beginning of text). Evaluations show both methods help reduce the success of attacks on models.

**Claims And Evidence:**

The authors perform a number of fairly controlled experiments to back up and test their hypotheses around what is causing issues in model role-learning, and they evaluate their proposed fixes across a few different attack strategies. This makes me reasonably confident that their claims are valid.

**Essential References Not Discussed:**

No

**Experimental Designs Or Analyses:**

The experiments and results shown in figures 2 and 3 isolating the effect of inserting sequences, instructions, or shifting tokens are fairly convincing and show fairly clear trends. The experiments generally are well-controlled and fairly targeted in identifying and testing the weaknesses and fixes proposed in the paper.
The evaluation for checking if PFT reduces general performance does not seem particularly comprehensive: it would be more useful to see if general evaluations such as GSM8k, AlpacaEval, etc. are reduced by performing this style of position-id shifted training, as the password task may be quite easy to fit to.

**Methods And Evaluation Criteria:**

The benchmarks evaluated on make sense to use, and the techniques are evaluated on both Llama and Gemma models. The closed-domain setting used (where user tokens cannot include instructions) is definitely a large simplification from real-world settings, but this is not unreasonable for the studies performed in the paper. It would be interesting to see if the methods propose harm or help performance in such open-domain settings.

**Other Comments Or Suggestions:**

- Figure 3 caption typo: “privildged” -> “privileged”
- 6.1 typo “ x②” -> “ ②”

**Other Strengths And Weaknesses:**

Overall, I think this paper is fairly strong, with a solid methodological setup that clearly establishes the proposed issues, and strong results that the fixes proposed do indeed improve these issues. Its largest weakness is the fact that only the closed-domain setting is examined, which feels like a fairly restrictive setup (and limits the scope of the insights in this paper) – I feel that in many cases users will be allowed to put in their own instructions and we wish for the model to follow these (to some reasonable extent). However, the paper is aware of the limitation and still makes novel and interesting insights.

**Questions For Authors:**

See my questions and concerns above, in particular with regards to open-domain performance.

**Relation To Broader Scientific Literature:**

This present a fairly clear identification and fix for two issues with LM safety, being novel in both identifying them and then proposing fixes.

**Theoretical Claims:**

This is a primarily empirical work, and the mathematical explanations where present seem correct.

---

> ### Author Rebuttal · Authors · 2025-03-31
>
> We are glad you like this work!
>
> **Open-domain performance** Thank you for your positive feedback! We choose a close-domain setting as it is easy to validate and evaluate the model role-separation capability. This is also a common choice in many security-related works, such as [StruQ](https://arxiv.org/pdf/2402.06363) . We agree with you that evaluating in open-domain would have a more comprehensive evaluation, and we would like to discuss it in future works. Thanks again for your support!
>
> **Evaluation of PFT impacts on other tasks**  we show PFT doesn’t incur extra performance cost compared to SFT on Alpaca dataset (sec 6.2).
>
> **Typos** Thanks for your careful reading! We will correct those typos and acknowledge your feedback in the next version.

---

> > ### Comment · Reviewer_r6iR · 2025-04-03
> >
> > Thank you for the response! I've read it and the other reviews and am keeping my score.

---

### Official Review · Reviewer_7NiM · 2025-03-24

**Overall Recommendation:** 4

**Summary:**

The paper studies how well LLMs are able to distinguish between different input roles like system, user etc. The authors motivate their work by claiming that existing fine tuning approaches do not teach the LLM genuine role differentiation but rely on spurious patterns picked up by the model during training. To this end, they propose an experimental framework where they use benign training data and adversarial test data to detect if the model really learns to distinguish roles or just memorizes patterns. They also discover shortcuts that arise from model behavior and show that minor perturbations can be used to cause model failure. They propose PFT, a fine tuning method that manipulates postion IDs to create numerical gap between system and user tokens. They show through experimental results that the PFT method is able to perform better than standard SFT models and other comparison models on Adversarial as well as Ordinary datasets.

## update after rebuttal
The authors discuss all questions and addressed the points raised by me in their rebuttal, so changing my score. Good work and good Luck!

**Claims And Evidence:**

The overall paper was a good read and many claims made were supported by experimental exploration and evidence. It  would be great if following points are strengthened by the authors,

1. Generalization of PFT to different prompt structures might need to be further tested beyond adversarial test examples, examples include longer prompts, mixed role messages etc.

2. Explain why delimiter tokens under perform, more detailed comparison across variants or exploration of the embedding space might strengthen this claim

3. Comparison and analysis of PFT performance to more related methods that perform similar encoding related fine tuning might help the readers understand the reasons for improvement and better understand the novelty

**Essential References Not Discussed:**

The paper has cited most of the related recent works, the following few recent papers might be of interest to the reviewers to consider citing:

ALIS: Aligned LLM Instruction Security Strategy for Unsafe Input Prompt(https://aclanthology.org/2025.coling-main.613/) (Song et al., COLING 2025)

Zverev, Egor et al. “ASIDE: Architectural Separation of Instructions and Data in Language Models.” (2025).

**Experimental Designs Or Analyses:**

The paper has good experimental design and studies. The controlled evaluation framework, short cut diagnosis and the quantitative evaluations are good. The paper might benefit from broader generalization testing and human centered evaluation to fully validate some of the robustness claims, but the overall experiment settings and empirical results are good.

**Methods And Evaluation Criteria:**

The paper discusses an important problem of role separation vs. pattern memorization which is very insightful and important. The controlled evaluation setup to train on benign data and test on adversarial data are methodologically sound and will help isolate the phenomenon. The following points are my feedback:

1. Could you please add more information about the datasets being used for training and testing, like size and how it was constructed? As 2K examples for the training set seems to be small and its not clear how different the initial and symm versions are.

2. The PFT introduces a numerical gap between the different role contents and is motivated empirically and intuitively by observing failure models, it would be good if there can be formal theoretical or ablation analysis as to why the proposed algorithm might result in better role separation understanding

**Other Comments Or Suggestions:**

None

**Other Strengths And Weaknesses:**

Strengths:

1. Clear problem framing - paper articulates the problem of role separation in LLMs in a focused and compelling way
2. Strong Empirical studies and analysis - authors design a thoughtful experimental framework that isolates shortcut learning from genuine role understanding
3. Well written and clear structure - paper is generally well-written and clearly structured


Weakness:
1. Limited Motivation for PFT - paper does not provide theoretical justification for the PFT method, nor does it deeply analyze why positional encodings work better than delimiter tokens for role separation
2. Limited Novelty - The PFT which shifts position IDs to encode roles is an adaptation of existing ideas from long-context learning and positional interpolation

**Questions For Authors:**

1. Could you please add more information about the datasets being used for training and testing, like size and how it was constructed? As 2K examples for the training set seems to be small and its not clear how different the initial and symm versions are.

2. The PFT introduces a numerical gap between the different role contents and is motivated empirically and intuitively by observing failure models, it would be good if there can be formal theoretical analysis as to why the proposed algorithm might result in better role separation understanding

**Relation To Broader Scientific Literature:**

The paper discusses role separation in LLMs, and is well situated within the broader literature on prompt injection, role conditioning and positional encoding in LLMs. It extends the work on role-specific embeddings (Wu et. al., 2024) by offering an alternative method by modifying position IDs. It also discusses about prompt injection attacks (Willson et. al, 2022; Yu et. al., 2023) and shits the focus to robust role separation rather than focussing on performance against known attacks.

References
Yu, J., Wu, Y., Shu, D., Jin, M., and Xing, X. Assessing prompt injection risks in 200+ custom gpts. arXiv
preprint arXiv:2311.11538, 2023.

Wu, T., Zhang, S., Song, K., Xu, S., Zhao, S., Agrawal, R., Indurthi, S. R., Xiang, C., Mittal, P., and Zhou,
W. Instructional segment embedding: Improving llm safety with instruction hierarchy. arXiv preprint
arXiv:2410.09102, 2024.

Willison, S. Prompt injection attacks against GPT-3, 2022. URL https://simonwillison.net/2022/Sep/
12/prompt-injection/.

**Theoretical Claims:**

The paper is majorly empirical and experimental with intuitive hypothesis and good experiments. As such, no issues with proof correctness arise. However, PFT introduces a numerical gap between the different role contents, it would be good if there can be formal theoretical analysis as to why the proposed algorithm might result in better role separation understanding

---

> ### Author Rebuttal · Authors · 2025-03-31
>
> Thank you for your thoughtful review and suggestions! We are glad that you find the paper well written, and the empirical evidence strong.
>
> Many of your questions about theoretical results and other embedding-based methods are great suggestions! We will study them in the future projects. More specifically,
>
>
> **Theoretical analysis**  Intuitively PFT introduces invariant signals to the SFT data and helps curb models learn various shortcuts. It would be interesting to study what property of the transformer architecture or the chat format explains the empirical phenomenon. We will leave it as future work, and will add more discussions in the next version.
>
> **Applying PFT to generalized prompt structures and embedding based methods** In related work, we acknowledged that the PFT in the current form doesn’t directly apply to generalized prompt structures, and embedding based methods (the one we cited is concurrent to our work) have the same motivations and could be used to enhance role-separation (in fact, our methods can be understood as an embedding-based methods — it changes the positional encoding and thus effectively changes the embeddings at each layer, while not requiring explicit embedding tuning). However, the main contribution in this paper is the clear definition of the role-separation problem, and controlled experiments for evaluation. It’s a natural next step to systematically study how to best incorporate role information at token level.  We also thank you for adding more related work. They are concurrent to this work, and we will definitely include it in the next version.
>
> **Why enhancing delimiter token doesn’t work well** We suspect it’s because the differentiating signal is still not strong enough, and a more robust approach is to manipulate tokenwise signatures (like PFT, or embedding-based approaches). We agree a deep theoretical analysis could formalize these intuitions.
>
>
> **Relationship to long-context learning** We briefly discussed the long-context learning works in line 427. In fact, the similar methods in long-context learning have completely different motivations (change positional encoding to simulate longer contexts in training), and we are glad that methods with completely different motivations could work! This suggests the big potential for more advanced techniques of manipulating positional encoding. We thank the reviewer for bringing this to our attention. We already briefly discuss them in the related work, but will add more discussion in the next version
>
> **Ablation studies** since we do controlled experiments (changing one component at a time), we are effectively doing ablation studies.
>
>
> **Dataset details** For dataset_initial, we discuss the main design and leave the details in Appendix C. We also include actual training data in the supplementary material. We introduce dataset_symm in Sec 4.1 line 206 as a way to combat short 0 by data augmentation. It’s also in the supplementary material. We agree that the different pieces of dataset info are introduced with the development of the paper, and thus are scattered around. In the next version we will add a more concentrated paragraph for data setup.
>
> **Whether the size of training data is too small** we think it really depends on the task, and empirically we do find this is more than enough for the model to learn role-separation (we run for one epoch and perform early stopping)

---

### Official Review · Reviewer_hqgQ · 2025-03-25

**Overall Recommendation:** 3

**Summary:**

The paper introduces the concept of role-separation learning, which reflects the LLM capability to distinguish the system instructions and user queries. The authors evaluate the role-separation capability of LLMs through a controlled experimental framework and conclude current fine-tuned models use task type exploitation and proximity to begin-of-text for role identification, which is considered as relying on superficial proxies.  The authors also propose the data-augmentation and Position-enhanced fine-tuning method, which is based on modifying position IDs, to achieve robust actual role-separation capabilities.

**Claims And Evidence:**

Yes

**Essential References Not Discussed:**

No

**Experimental Designs Or Analyses:**

I checked three main experimental designs and all designs make sense to me:
1.  Using accuracy on constructed tasks to present the model role-separation capability and compare the performance of different fine-tuning methods. The input question consists of a clear system instruction and ambiguous user query, which could mislead the LLM to generate an undesired response. When the LLM provides the answer following the provided system instructions under some misleading attacks, the LLM presents role-separation capability. The paper designs different fine-tuning datasets and compares their performance through Table 1 and Table 2.
2. Evaluating the impact of non-essential information like "You are an AI assistant" with different numbers and insert positions. The non-essential instruction could be inserted before or after the key instruction. When the non-essential instruction is inserted after the key instruction, increasing the number of non-essential instructions shifts the position of the key instruction backward.

3. Evaluating the impact of positional distance of the key instruction. Adjust the distance by inserting empty tokens before the key instruction or shifting the position IDs of the key instruction. This experiment does not provide general instructions, therefore isolating the impact of the key instruction positions.

**Methods And Evaluation Criteria:**

Yes

**Other Comments Or Suggestions:**

Some examples from the selected datasets would help the readers understand the problem setup and challenges more easily.

**Other Strengths And Weaknesses:**

Strengths:
1. The paper investigates an important concept of role separation since recent real-world applications of LLM increasingly incorporate instructions and information from multiple sources.

2. The paper evaluates and improves the actual role-separation capability in isolation from pattern-matching behavior and potential shortcuts. The paper evaluates the performance against multiple types of attacks.

3. The author proposed the Position-enhanced fine-tuning method which has statistically better performance in improving role-separation capability compared to vanilla SFT.

Weaknesses:
1. Evaluation of the framework and method using more models could be helpful. Models with different scales and types present different capabilities of comprehending, reasoning, and instruction-following. Experiments on different models could provide a more robust and comprehensive view of the proposed method. Additionally,  for the PFT method, as shown in Sec 6.1, the optimal distance d may depend on the model type.  Conducting experiments on a wider range of models and providing corresponding hyperparameter d will facilitate the further application of the proposed method.

2. It’s unclear whether “Accuracy” is a precise metric to evaluate the role-separation learning capability. The role-separation capability requires the LLM to correctly identify the task, corresponding solving strategy, and problem. However, accuracy needs the LLM to correctly understand the task and solve it, thus is also influenced by the model problem-solving capability.

3. Some related work section needs more details. For instance, in the Prompt Injection Attacks section, how the prompt injection attacks are designed and related to this work. It is also unclear why the mentioned online game datasets are suitable for this work.

**Questions For Authors:**

1. I’m curious about whether providing some similar in-context examples will improve role-separation capability.
2.  In Sec 4.1, the authors mention the task-type association shortcut influences the accuracy, does it mean the second and third columns of numbers shown in Table 1 also include the contributions of this shortcut to the task success? If so, is there any better way to distinguish the contribution of this shortcut and actual role-separation capability?

**Relation To Broader Scientific Literature:**

Existing works evaluating role separation capability do not decompose the influence of role separation capability and other confounding factors. This work demonstrates that LLM performance also depends on pattern matching and superficial shortcuts. This work designs a controlled experiment framework to isolate the role separation capability from pattern memorization. And design experiments to estimate the influence of task-type association shortcuts and proximities to begin-of-text shortcuts. This work tries to evaluate the actual role separation capability which precisely reflects the LLM capability to distinguish systems instructions and user queries.

**Theoretical Claims:**

There is no theoretical claim demonstrated in the paper.

---

> ### Author Rebuttal · Authors · 2025-03-31
>
> Thank you for your review! We are glad you find the problem of role-separation important, and the experiment designs make sense.
>
> For your questions about whether other factors (such as model capability or evaluation metrics) confound the conclusions, we discussed in the paper that controlled experiments help remove them. But we will add more clarifications to the next version. To your questions, more specifically:
>
> **Regarding evaluation with more models** Yes, we agree and include results on both Llama and Gemma models, and find common trends. We want to emphasize that the two models are very different. In particular, Llama 3 models use a large RoPE base frequency hyperparameter (500k) to support long contexts, whereas Gemma uses only 10k. This could partially explain the different choices of $d$ in PFT.  But more importantly, we use controlled experiments to further remove the confounding of other model capabilities (comprehending, reasoning, etc): in each experiment, we change only the targeted aspect while holding other aspects the same; then we use the performance difference/trend as evidence for each claim.
>
> **On use of accuracy as metric for role-separation capabilities** First, the tasks are chosen to be simple so that the model's problem solving skill is not a limiting factor (the model performs well in ordinary data).  Second, even if model capabilities might still affect the absolute metric scores, we can remove this confounding effect by looking at differences in accuracy. This is what we did: from side by side comparison in table 1, to trend analysis in Fig 5, we use the accuracy difference to prove our claims.
>
>
> For other questions:
>
> **On why prompt injection attacks relate to role-sep problem** we used prompt injection attacks as motivation for the role-separation problem (section 2), and discussed that we used these adversarial datasets (many are collected from online games) to test the model OOD performance as true measure of role-sep capability (Sec 3). We discussed those prompt injection datasets in the paragraph of line 148, and used Fig 1 as illustration of their designs. We acknowledged that descriptions of the prompt injection attacks are scattered (because of page limits). In the next version, we will add a more concentrated paragraph discussing adversarial datasets.
>
> **On in-context learning**  We had early experiments with ICL, and found it suffers the same task-type association problem as SFT. Because ICL and SFT share similar characteristics, we stick with SFT for this paper. It would be an interesting future direction to explore how our results generalize with ICL.
>
> **On question about shortcut and Table 1** Yes, the “good” result of the second column is an illusion caused by task-type association, and we find it by swapping the contents of the system and the user (line 195).  Then we remove the task-type spurious correlation by training on data augmented with the swapped examples, and observe extra performance jumps in the 3rd column of table 1 (line 206).

---

### Decision · Program_Chairs · 2025-05-01

**Decision:**

Accept (poster)

**Comment:**

The paper makes valuable contributions through its well-framed investigation of how LLMs distinguish input roles, compelling empirical studies isolating shortcut learning, and the promising Position-enhanced Fine-Tuning (PFT) method. Suggestions for improvement include: strengthening the theoretical justification for PFT (7NiM), expanding evaluation beyond closed-domain settings (r6iR), testing across more diverse prompt structures and models (7NiM, hqgQ), providing more detailed delimiter token comparisons, and refining the "Accuracy" metric (hqgQ). While the authors addressed many concerns in their rebuttal, more thoroughly incorporating these refinements would further enhance this valuable contribution to understanding role separation in LLMs.